# Influence of COVID-19 Pandemic Uncertainty in Negative Emotional States and Resilience as Mediators against Suicide Ideation, Drug Addiction and Alcoholism

**DOI:** 10.3390/ijerph182412891

**Published:** 2021-12-07

**Authors:** Blanca Rosa García-Rivera, Jorge Luis García-Alcaraz, Ignacio Alejandro Mendoza-Martínez, Jesús Everardo Olguin-Tiznado, Pedro García-Alcaráz, Mónica Fernanda Aranibar, Claudia Camargo-Wilson

**Affiliations:** 1Faculty of Administrative and Social Sciences, Universidad Autónoma de Baja California, Ensenada 22890, BC, Mexico; maranibar@uabc.edu.mx; 2Department of Industrial Engineering and Manufacturing, Autonomous University of Ciudad Juarez, Ciudad Juarez 32310, CHI, Mexico; 3Department of Graduate Studies, Universidad Anáhuac, Anáhuac 01840, DF, Mexico; alejandro.mendozam@anahuac.mx; 4Faculty of Engineering, Architecture and Design, Universidad Autónoma de Baja California, Ensenada 22860, BC, Mexico; jeol79@uabc.edu.mx (J.E.O.-T.); ccamargo@uabc.edu.mx (C.C.-W.); 5Agricultural Technology High School 148 (CBTA), Comala 28950, CL, Mexico; pedrogarcia148@dgetaycm.sems.gob.mx

**Keywords:** pandemics, uncertainty, emotional states, resilience, suicide ideation, drug addiction, alcoholism, faculty students

## Abstract

This research uses structural equation modeling to determine the influence of uncertainty due to the COVID-19 pandemic as an independent variable in the negative emotional states and resilience (as mediating variables) vs. drug addiction, alcoholism, and suicide ideation as dependent variables in 5557 students from a public state university in Northern Mexico. The five variables are related through eight hypotheses and tested using partial least squares. We used an adapted questionnaire sent by email in May 2020. Findings show that uncertainty facing the COVID-19 pandemic had a direct and significant influence on negative emotional states and a significant inverse effect on resilience; in the trajectory, drug addiction and alcoholism, and suicide ideation are explained.

## 1. Introduction

The COVID-19 (Coronavirus outbreak 2019) causes severe acute respiratory syndrome type-2 (SARS-CoV-2) and has become a public threat to humanity [1]. News stating that the virus was spreading worldwide from Wuhan—Hubei province in China—began spreading in December 2019. The virus affected patients with severe dyspnea and pneumonia [2]. In early 2020, almost 1000 confirmed cases of this infection were reported by the Chinese disease control center [3]. The cases had non-specific signs of the disease, mainly characterized by dyspnea and pneumonia. Today, there have been millions of confirmed cases and deaths worldwide in what appears to be the most significant public health emergency of humanity in modern times [3].

In March 2020, the WHO declared a pandemic and called on all countries to take emergency measures. In Mexico, the virus began to appear as of April 2020. The health care system declared quarantine immediately. In the beginning, it was expected that by the end of the first half of 2020, “normal activity” would be risk-free. However, conditions have remained on red alert. Although cases had declined after a large portion of the population received the vaccine, new waves arise, keeping the world continuously alert today [4].

This situation has received massive attention from the media, leading to them having the highest ratings [5]. At time of writing (29 August 2021), a search in Scopus shows 137,969 results in title and 191,045 results in the abstract, title, or keyword regarding the word COVID-19. However, most of the published information have been guides, manuals, and clinical reports, and very few technical and scientific papers focus on e-learning and distance education.

The study of psychosocial risks within students during this pandemic has received attention from scholars and professionals in the area [6,7,8]. There are several studies concerning psychological impact, mental health, and psychosocial risks. Most of these reviews have reported adverse psychological effects due to isolation [8,9]. Some of the observed effects are stress and anxiety due to the uncertainty of the length of the isolation period; fear of being infected; frustration and boredom; confusion and anger due to lack of adequate information, financial losses, and mismanagement of quarantine by civil authorities, overcrowding and domestic violence [6]. 

Quarantine due to the COVID-19 outbreak has been an unpleasant experience for most people [10]. The benefits of isolation must be later analyzed in greater detail since the psychological costs observed in recent studies are high [11,12,13]. However, there is an urgent need to analyze the consequences of psychosocial risks under the current isolation conditions that university students are experiencing, including increased drug addiction, alcoholism, and suicide ideation [14]. 

Theoretical findings identifying the behavior of the aforementioned psychosocial risk factors in emergencies such as the COVID-19 pandemic currently experienced are relevant. Moreover, from a practical point of view, there must be empirical evidence regarding student’s current perceptions when facing this situation. Socially, prevention should be used to avoid the situation escalating into significant problems, such as severe alcoholism, drug addiction, suicides, and emotional imbalances that put at risk the students’ lives due to being quarantined for such a long period. A methodological contribution of this research is the use of the structural question modeling where the incremental mediating role of negative emotional states is analyzed vs. the decremental mediation of resilience against COVID-19 towards drug addiction and alcoholism and suicide ideation. 

### 1.1. Hypotheses 

Research shows that COVID-19 pandemic uncertainty is observed as a mental health risk due to lockdown restrictions, the virus spread and related risk factors [15]. Life-threatening situations have psychological effects on both emotional and cognitive behaviors, influencing attitudes and beliefs. In addition, considering the economic impact of the pandemic on families and budgets, uncertainty became a strong mediating emotion facing this pandemic [16,17]. A better understanding of the biases they lead to could improve judgments and decisions in situations of uncertainty [18]. Due to the COVID-19 pandemic, there have been reports of possible collective trauma, causing global anxiety and heightened stress [19]. According to research, the COVID-19 pandemic is affecting the four basic pillars of resilience: Mental: awareness, adaptability, decision making, positive thinking; Physical: endurance, nutrition, recovery, and strength; Social: family, communications, connectedness, social support, and teamwork; Spiritual: core values, perseverance, perspective, and purpose [20,21]. Researchers indicated that approximately 41% of respondents were experiencing severe levels of anxiety—females being more anxious than males—and uncertainty related to their academic performance, completion of the current semester, exam dates, and isolation issues, the stronger the more related to uncertainty [22]. In addition, research showed that 61% of sampled students were experiencing anxiety due to uncertainty related to relatives infected, conflicts at home, noisy environments, and increased consumption of tobacco, drugs, and alcohol [23]. Recent studies showed that more than 20% of respondents had anxiety due to uncertainty in family income stability, living with parents and relatives infected with COVID-19, economic stressors and completion of the current semester [24]. Other research showed that more than 20% of the sampled students had mild to severe anxiety due to uncertainty in personal activities at home, adopting COVID-19 safety guidelines and an inability to cope with their problems [25]. 

### 1.2. Negative Emotional States

Social distancing conditions due to the pandemic have caused psychosocial effects due to uncertainty, feelings of threat, and confinement; consequently, emotional states such as anger, frustration, insomnia, stress, anxiety, and depression arise [6,24]. Being in a continuous state of concern and uncertainty results in states of anxiety and depression [26,27]. It is observed that in university students these depressive states are exhibited more by women than men [28,29]; therefore, it is of high importance to know the current state of university students and the impact that several factors have on their mental health, such as isolation and family pressure, violence, overcrowding, work and academic overload, individual characteristics, physical and space conditions, and the financial resources available to them. 

Due to this, it was decided to include potential predictors of emotional effects such as depression, anxiety, and stress in the sampled students Research has also shown that uncertainty has directly impacted emotional negative states in university students during the COVID-19 pandemic [20,30]. Recent studies showed that more than 50% of the sampled students had a decrease in psychological wellbeing due to the lockdown and isolation [31]. An increase in substance consumption as a coping strategy produced negative states such as depression in more than 80% of the students. This fact suggests an unparalleled growth in depression and anxiety experienced by the students due to prolonged unemployment, financial insecurity and family situations contributing to uncertainty. Given these previous empirical findings, we propose:

**Hypothesis** **1.***COVID-19 pandemic uncertainty has a strong, significant impact on negative emotional states*.

### 1.3. Resilience against COVID-19

Resilience is defined as the ability to recover from adversity when faced with a traumatic situation, a loss, or a catastrophe, and strengthening resources, competencies, and emotional connection after the experience [32]. It is also understood as a dynamic process that involves resisting, building, and self-affirming [33]. Resilience has been a topic of study for more than four decades since noting that children in hostile or highly-violent environments were able to develop characteristics of great strength and resistance, living an everyday life.

Faced with the current pandemic, many young people are trying to find meaning in their lives, which implies rebuilding and committing to a new dynamic [34]. Recent studies showed that more than 30% of students experiencing stress, anxiety, depression, and fear of infection improved their resilience by getting involved in various activities such as physical exercise, recreational activities (watching TV series, reading storybooks, online and offline gaming and household chores) that helped them to cope with the situation better. In addition, support from their family and friends increased their resilience as a protective factor, according to [23]. Moreover, avoiding media as a protective coping mechanism increased their resilience [25]. Given these previous empirical findings, we propose:

**Hypothesis** **2.***COVID-19 uncertainty has a direct, negative effect on resilience*. 

### 1.4. Suicidal Ideation

Suicidal ideation is a variable that has sparked international attention [35]. Nowadays, more than 800,000 people take their own lives each year [35,36]; specifically, in the 15–29 age group, suicide is the second cause of death worldwide [36]; the male gender being the most susceptible [37,38]. Previous research refers to the fact that the most common factors that lead university students to think about suicide include academic overloading, insufficient rest time, pressure of being alert to fulfill specific responsibilities, depressive and anxiety disorders, stress, and other psychosocial risk factors they face, as well as family problems, socio-economic limitations, substance, and alcohol abuse, among others [35].

Suicidal ideation has been defined as the recurrent thought and planning that an individual performs for committing suicide, but that it is not carried out [39]. There has been a consensus among scholars in the assertion that suicide ideation has several observed stages that start with a desire to die, followed by passive fantasies of suicide, leading to suicide ideation without a particular method [10].

In scientific literature, it is observed that in Latin American countries, including Mexico, suicide ideation and suicide attempt and completed suicide are less than those registered in Europe and in the United States. In general, in Latin America, the rate of suicide ideation among students from degrees other than Medicine fluctuates between 10% and 15%. This rate is higher for medical students, where it fluctuates between 17% and 22% [40].

Currently, due to isolation and uncertainty in the face of the pandemic, suicide ideation must be measured in order to see whether it has increased and whether there are risk profiles that must be observed and cared for [41]. Although there are numerous studies on suicide ideation, it is noted that in Mexico, there are very few; therefore, studies should be carried out to assess suicide ideation among students by analyzing variables such as gender, substance and alcohol abuse, other socio-demographic characteristics and mediating variables that explain why suicide ideation could be higher. Studies showed that higher than normal levels of somatization, obsessive-compulsive disorder, anxiety, phobic anxiety, paranoid and suicidal ideation, and general severity index were observed in sampled students during the COVID-19 pandemic [42]. Research shows that uncertainty relates directly to suicidal ideation, anxiety and obsessive-compulsive disorders [43]. Negative states promote suicidal ideation, according to [44]. Research shows that [45], coping strategies such as leisure time, work, exercising, and sleeping had a mediating effect increasing resilience and decreasing stress and distress; on the other hand, resilience directly negatively affects suicidal ideation [46,47]. Given these previous empirical findings, we propose:

**Hypothesis** **3.***COVID-19 pandemic uncertainty has a direct, positive, significant effect on suicidal ideation*. 

**Hypothesis** **4.***Negative emotional states due to COVID-19 pandemic uncertainty promote suicidal ideation*. 

**Hypothesis** **5.***Resilience against the COVID-19 pandemic has a direct, negative effect on suicidal ideation*.

### 1.5. Drug Addiction and Alcoholism

Moreover, substance use and increased tobacco consumption as a coping strategy are important risk factors for anxiety during COVID-19 times [48,49,50]. Alcohol and substance abuse compared to previous trends have been increasing [51]. Recent studies have shown that more than 30% of students in Mexico use alcohol socially, most between 19 and 25 years old. This problem affects students’ performance and mental and physical health, and leads to domestic violence and aggression among young people. In some cases, severe injuries have been reported [52]. The instrument used, AUDIT, was adapted and modified to integrate the dimension of drug addiction since this scale is for alcohol use disorders only. Most previous studies that used this instrument showed that early detection of alcohol and substance abuse allowed successful interventions and the implementation of prevention programs and policies aimed to reduce abuse [53,54,55,56]. Therefore, we can state that COVID-19 pandemic uncertainty promotes substance and alcoholism abuse [57]. Then we propose H6 as follows: 

**Hypothesis** **6.***COVID-19 pandemic uncertainty has a direct, positive effect on drug addiction and alcoholism in university students*. 

Previous research has also shown that negative emotional states promote higher drug addiction and alcoholism [58]. Then we propose H7: 

**Hypothesis** **7.***Negative emotional states directly and positively affect drug addiction and alcoholism in university students*.

Last, research has shown that resilience against the COVID-19 pandemic diminishes the use of drugs and alcohol [59]. Therefore, we state H8: 

**Hypothesis** **8.***Resilience against COVID-19 pandemic has a direct, negative effect on drug addiction and alcoholism in university students*. 

Figure 1 summarizes all the stated relationships between the stated variables and hypotheses. 

In Table 1 we present the expected influence of the exogenous variables related to the endogenous variables as proposed in the hypotheses presented.

Following, please find Table 2 where we express the hypotheses proposed in this research. As noticed, we propose eight hypotheses related to the variables of the Model.

## 2. Materials and Methods

### 2.1. Study Design

This study is a cross sectional analysis of data. Ex post facto, non-experimental, explanatory design that was conducted through an online survey. Structural equations with latent variables under the method of partial least squares [60] were used for the analysis.

### 2.2. Data Collection

The instrument was sent electronically in May 2020 to the total of students enrolled in bachelor’s degree, master’s degree and PhD degrees. The population of this campus is 10,975 students (UABC Ensenada campus, 2020-1), n = 5557, obtaining a response rate of approximately 50.63%. The information of the answered questionnaires was uploaded into a database that was edited and analyzed in the Statistical Package for the Social Sciences (IBM SPSS) version 23 for Windows, and the Smart PLS version 3.

### 2.3. Instruments

The instrument used for data collection included the following sections:Students questions regarding their sociodemographic characteristics including gender, marital status, age, scholarity, academic demographic information such as academic program and scholarship status to categorize the demographic variables.COVID-19 pandemic uncertainty: we developed this part using 6 questions to measure uncertainty as a unidimensional variable with a Likert scale of five points. Examples of the questions used are: “I was afraid when facing the pandemic”, “I felt that life is very fragile”, “I feared to be infected”, “I was terrified to imagine someone dear was going to die”.Resilience scale CD-RISC25 [61]. Focused on determining resilience. Resilience is the ability of human beings to adapt and overcome adverse situations, measured through their level of positive response to risk situations as a multidimensional construct. This instrument consists of 25 items that must be answered in a 5-point Likert scale (1 to 5). It consists of five dimensions: persistence/tenacity/self-efficacy (items 10–12, 16, 17, 23–25); control under pressure (items 6, 7, 14, 15, 18, 19, 20); adaptability and support networks (items 1, 2, 4, 5, 8); control and meaning (items 13, 21, 22) and spirituality (items 3, 9).DASS-21 questionnaire short version (anxiety scale [62]). This scale has six states that measure what the respondent is “feeling at this moment.”Adaptation of the Scale for Suicide Ideation. Focused on determining the suicidal ideas of students in the face of the pandemic. It is based on the Scale for Suicide Ideation (SSI-W) and it consists of six questions that must be answered in a range of one to two [63].Adaptation of the Scale for Disorders by Drug Addiction and Alcoholism. Focused on determining trends in alcohol and drug use. It is based on the scale Alcohol Use Disorders Identification Test (AUDIT) proposed by Saunders et al., 1993 [64] and it consists of 8 items that must be answered on a scale of zero to three.

### 2.4. Statistical Analyses

Information was generated about the descriptive statistics of subscales, including: mean score, standard deviation, as well as the Pearson moment-product bivariate correlations. With the aim of analyzing the significant differences of the mean scores of students by sex (men vs. women), the Student’s T-test was necessary, supplemented by the Levene’s test.

## 3. Results

### 3.1. Unit of Analysis

Research subjects who collaborated in this project were students from a public state university, P = 65,000; n = 5557. Their socio-demographic data were: Gender 63.9% (f = 3551) Women, 36.1% (f = 2006) Men; Marital status 90.6% (f = 5036) Single, 4.7% (f = 259) Civil union, 3.7% (f = 210) Married, 1% (f = 52) Other; Scholarship 93.6% (f = 5201) Bachelor’s degree, 3.7% (f = 203) Specialization, 1.7% (f = 105) Master’s degree, 1(f = 48) Other; Age, 71% (f = 3970) aged 17 to 22; 21% (f = 1173) 23–27 years old; 8% (f = 414) 28 to more than 63. All students received detailed information regarding the purpose of the study and provided online informed consent to participate in the study. The survey was completed anonymously to ensure the confidentiality and reliability of the data. The study targeted the entire population and the response percentage was close to 10%. 

### 3.2. Contrast of the Hypotheses

The statistical analysis used for the contrast of the hypotheses was based on the understanding of the nature of the question and the research hypothesis. It was necessary to use a Structural Equations Modeling with latent variables under the method of partial least squares (SEM–PLS) using the Smart PLS software. Several multiple regression analyses were generated parallelly as well as their respective standardized beta coefficients.

In order to assess the validity of the instruments, the use of the Factor Loading Analysis was necessary, complemented with the analysis of the Average Variance Extracted (AVE) and the Discriminant Validity (Appendix A). For the analysis of the reliability of the instrument, the coefficients of Cronbach’s alpha, Rho A and Composite Reliability (CR) were obtained. The Pearson product-moment coefficients were calculated. The structural modeling (SEM–PLS) was developed under the theoretical foundations and the reflective method.

The Structural Equation Modeling (SEM–PLS) allowed for the visualization of the exogenous variable (E3 Uncertainty in the face of the COVID-19 pandemic) in mediating variables E14 Resilience against COVID-19); as well as endogenous variables (E12 Drug addiction and E11 Suicide ideation) with the corresponding items that integrate it, thus jointly evaluating these hypotheses.

### 3.3. Reliability and Validity

Table 3 shows the results of the descriptive statistics (mean, standard deviation, Cronbach’s alpha reliability coefficients [Fluctuation, Min = 0.73, Max = 0.85], Rho A [Fluctuation, Min = 0.74, Max = 0.86], Composite reliability [Fluctuation, Min = 0.85, Max = 0.90]). This allows the assertion that the instruments have very good reliability levels, having internal consistency in their results [65].

Regarding validity, the Average Variance Extracted (AVE) was (Fluctuation of a Min = 0.65 to a Max = 0.72), meanwhile the Discriminant coefficients (Square root of AVE) were (Fluctuation of a Min = 0.81 to a Max = 0.85). This confirms the validity of the instruments at the optimal level, affirming that the instruments measure what they intend to do so.

Table 3 shows the results of Descriptive statistics, Reliability, Validity, and Correlations between the subscales of the Structural equations modeling of trajectories with latent variables.

### 3.4. Correlations between the Subscales of the Instrument

Table 3 shows Pearson product-moment bivariate correlations as follows: significant direct correlations were reported between E10 Negative emotional states with E11 Suicide ideation (r = 0.306) and E12 Drug addiction and alcoholism (r = 0.130), with E3 Uncertainty in the face of the pandemic (r = 0.461). Significant direct correlations were reported between E11 Suicide ideation with E12 Drug addiction and alcoholism (r = 0.264), with E3 Uncertainty in the face of the pandemic (r = 0.254). In the same way, a significant direct correlation was obtained between E12 Drug addiction and alcoholism (r = 0.117). However, significant inverse correlations were obtained with E14 Resilience against COVID-19 with all the subscales of the instrument: with E10 Negative emotional states (r = −0.283), with E11 Suicide ideation (r = −0.336), E12 Drug addiction and alcoholism (r = −0.111), with E3 Uncertainty in the face of the pandemic (r = −0.127).

### 3.5. Significant Differences between Resilience and the Subscales of Suicide Ideation, Emotional States, and Drug Addiction and Alcoholism Regarding Students’ Sex

In order to analyze the significant differences between Resilience against COVID-19 and the subscales of Emotional states, Suicide ideation, and Drug addiction and alcoholism between men and women, some statistical tests of mean differences were performed with the Student’s *t*-Test and Levene’s test; these are shown in Table 4 and Table 5.

Table 4 and Table 5 show the significant differences in the 5 subscales of the study, except in E11 Suicide ideation, where there were no significant differences reported. In the E10 Negative emotional states, women obtained a higher mean score (mean = 2.62) vs. men (mean = 2.38); in E12 Drug addiction and alcoholism, men obtained higher mean score (mean = 1.12) vs. women (1.08); in E14 Resilience against COVID-19, men had a higher mean score (mean = 3.71) vs. women (3.63); whereas in E3 Uncertainty against the pandemic, women obtained a greater mean score (mean = 2.49) vs. men (mean = 2.11)

Following, Figure 2 shows the structural equations Model with latent variables that describe the hypotheses results.

### 3.6. Explained Variance of Subscales in the SEM PLS

Uncertainty in the face of COVID-19 influenced in a direct significant way from its standardized beta coefficient (0.469) in E10 Negative emotional states and explained approximately 22% of its variance based on its R squared. On the other hand, E10 influenced in an inverse significant way from its standardized beta coefficient (−0.131) in E14 Resilience against COVID-19, and explained approximately 1% of its variance based on its R squared.

Following the trajectories, E12 Drug addiction and alcoholism was influenced by the standardized beta coefficients of E10 Negative emotional states (0.070), by E3 Uncertainty in the face of the pandemic (0.076), in an inverse significant way, by E14 Resilience against COVID-19 (−0.083); these subscales explain approximately 1% of E12 Drug addiction and alcoholism.

On the other hand, following the trajectories, E11 Suicide ideation was influenced by the standardized beta coefficients of E10 Negative emotional states (0.152), by E3 Uncertainty in the face of the pandemic (0.150), in an inverse significant way, by E14 Resilience against COVID-19 (−0.269); these subscales explain approximately 17% of E11 Suicide ideation.

It is important to observe that the explanation of E12 Drug addiction and alcoholism by the exogenous variable E3 Uncertainty in the face of the pandemic and the mediators such as E10 Negative emotional states and E14 Resilience against COVID-19 was very small to explain less than 1% of its variance; even though it was significant, it was not relevant and it was eliminated from the analysis. For the above reasons, the SEM was run again, eliminating E12 Drug addiction and alcoholism from the model. This is shown in Figure 3.

In order to corroborate the model, the bootstrapping was necessarily run again and it is displayed in Table 6.

Table 6 shows that the SEM PLS Model and its subscales confirm the specific hypotheses with a 95% confidence interval. Table 7 embeds a summary of results of the final SEM Model.

### 3.7. Mediation Analysis

We calculated the specific values of total, indirect and direct effects, and cumulative explained variance with Bootstrapping use in 5000 sampling. 

### 3.8. Increasing Mediation

Increasing mediation in the SEM was integrated in Hypotheses H1, H2 and H3. 

Specific values of total, indirect and direct effects, standard deviation, significant T values and explained variance are shown in Table 8.

According to the results of Table 8 and VAF, we confirm that 41% of E3 uncertainty effect on suicidal ideation E11 can be explained by negative emotional feelings, E10 as moderating significant increasing variable. 

### 3.9. Decreasing Mediation

Decreasing mediation in the SEM was integrated in Hypotheses H1, H2 and H3: specific values of total, indirect and direct effects, standard deviation, significant T values and explained variance are shown in Table 9.

According to the results of Table 9 and VAF, we confirm that 15% of E3 uncertainty effect on suicidal ideation E11 can be explained by Resilience E14 as a moderating significant decreasing variable on E11. 

Nevertheless, according to [60], partial mediation can be proved when VAF values exceed 0.2 and its totally mediating when these values exceed 0.8. in this case, E10 is confirmed as a mediating variable explaining 41% of E11 increasing suicidal ideation. On the other hand, E14 cannot be used as a mediating variable decreasing E11 due to the fact that it only reduces 20% of E11.

## 4. Discussion

This study determined that uncertainty in the pandemic among researched students was influenced directly and significantly by negative emotional states, resilience against COVID-19, drug addiction and alcoholism, and suicide ideation. Moreover, it showed that negative emotional states are determined as a significant, partial effect having a 41% impact on suicidal ideation. Nevertheless, resilience against COVID-19 is not determined as a significant, partial, decremental mediating effect, having only a 15% impact on suicidal ideation. However, it was significant, but not enough to consider it as a mediating partial significant variable. On the other hand, the drug addiction and alcoholism relationship based on the R squared is significant but very low, so it was removed from the model; therefore, other factors not included in the study need to explain these variables. Resilience and negative emotional states were examined as possible mediators between COVID-19 uncertainty and suicidal ideation, drug addiction, and alcoholism in a sample of faculty students during the pandemic, but were not enough to be considered as mediating variables. However, resilience was found to play a significant role in the transmission of uncertainty’s impact on suicide ideation. Research shows that people with character strengths such as optimism and resilience are better equipped to deal with difficult situations [66]. However, the model showed that the mediating effect of the studied variables on drug and alcohol abuse was very weak

On the other hand, the explanation of the suicide ideation based on the R squared is significant, reaching approximately 17% of its variance. Suicide ideation due to COVID-19 based on modeling the variable’s questions shows essential findings from a prevention posture since negative thoughts can be “detected” in students. In addition, we observe that in the negative emotional states variable, women obtained higher scores compared to men, and no significant differences were observed in suicide ideation between men and women. Moreover, drug addiction and alcoholism show higher mean scores in men compared to women. In addition, resilience against COVID-19 shows higher mean scores in men compared to women. However, the uncertainty of the pandemic shows that women reported higher mean scores compared to men.

The instrument used in this research obtained adequate levels of reliability by Cronbach’s alpha coefficients, Rho, and composite reliability. Regarding validity, adequate levels were obtained by explained average variance (AVE), factor loadings, and discriminant validity. 

As explained in the first section of this research, the number of suicides is not related to suicidal ideation. However, most suicides in the Americas region occurred from ages 25–44 (36.8%) and 45–59 (25.6%). Only 19.9% of the suicides occurred in ages 60 or older, and ages over 70 had a suicide rate of 12.4 per 100,000—the highest of all the age groups in the Americas [67]. Every committed suicide started with suicidal ideation; hence, the role of prevention is crucial. Prevention must be a priority related to the mental health of students that can lead to suicidal intent. Young Males and the elderly continue to be at higher risk. It is recommended to evaluate mental health systems, existing legal frameworks, and the availability of programs, services, and resources to adequately prevent and treat problems associated with suicidal behavior and suicidal intent; family involvement and a structured psychosocial intervention (on the principles of acceptance and commitment therapy and cognitive behavioral therapy) to help participants in making associations between harmful alcohol use and suicidal ideation is important [6,68]. The length of the COVID-19 pandemic and lockdown, stay-at-home orders, the increased social isolation, restricted outdoor activities, and the challenges of online classes have negatively impacted higher education. The findings of our study emphasize the urgent need to develop interventions such as psychoeducation, supportive counseling, assertive community treatment, and physical activity strategies—group-based, delivered face-to-face or distally—to address the mental health of students, addictions, alcohol abuse, and suicidal intent [69].

Previous studies show that addictions, uncertainty, and suicidal intent, frequently expressed by our participants, have been shown to affect students’ [70] adversely; previous research also shows correlations to increased stress and mental health [71]. This fact is similar to recent findings of deteriorated mental health status among Chinese students [24] and increased internet search queries on negative thoughts in the United States [72]. The findings on the impact of the pandemic on sleeping and eating habits are also a cause for concern, as these variables have known correlations with depressive symptoms and Anxiety [73]. Hence, our results confirm previous evidence.

### 4.1. Limitations to this Research

Surprising results were given as numerous students were answering the questionnaire very quickly. They also asked for help since they felt scared and alone when coping with the pandemic’s uncertainty. Unfortunately, we were unable to offer intervention in this first stage of the study. Recommendations were made to the faculties to pay special attention to students asking for help during this period. One limitation is having transversal research in one moment only, not comparing the evolution of the states and resilience at different moments of this pandemic. 

### 4.2. Implications and Opportunities for Further Research

The current context of the COVID-19 pandemic opens several areas of research focused on adaptive mechanisms that students are learning. It is essential to compare the uncertainty and resilience that students have developed as part of their adaptation and how these variables impact their coping mechanisms and their general response to the pandemic. 

In addition, a second application of the instrument is recommended to compare the evolution of the variables over time. 

## Figures and Tables

**Figure 1 ijerph-18-12891-f001:**
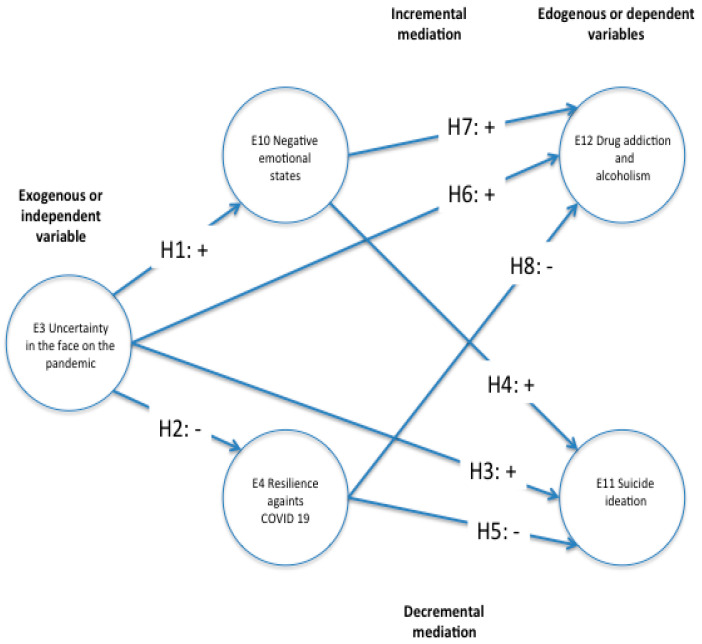
Structural equations model with latent variables of the hypothesis testing. Source: Own elaboration.

**Figure 2 ijerph-18-12891-f002:**
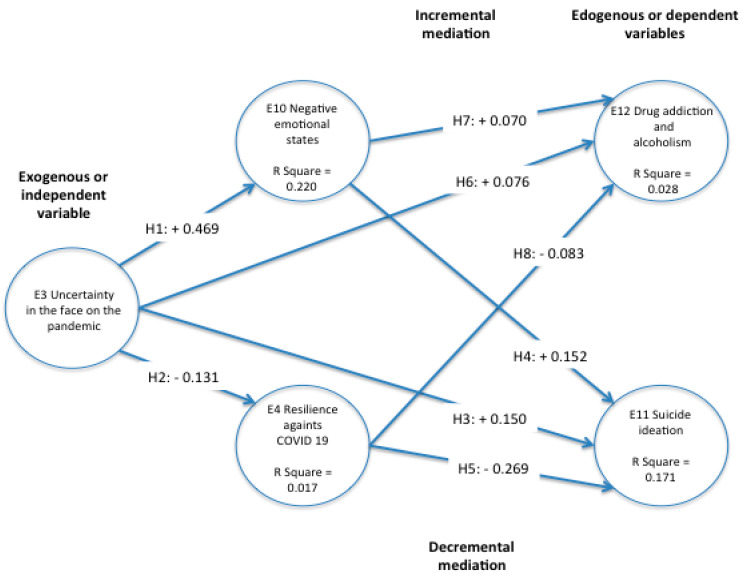
Model of structural equations with latent variables SEM-PLS. Source: own elaboration.

**Figure 3 ijerph-18-12891-f003:**
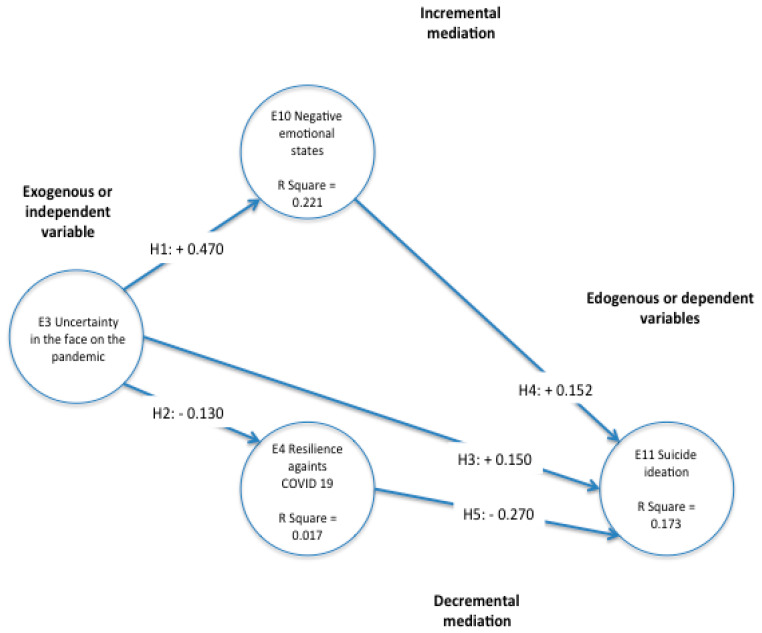
Model of structural equations with latent variables SEM-PLS. Source: own elaboration.

**Table 1 ijerph-18-12891-t001:** Expected influence of the independent variable (exogenous) in connection with the dependent variables (endogenous).

Hypothesis	Exogenous Variables	Influence	Sign	Endogenous Variables
1	E3 Uncertainty in the face of the pandemic	====˃˃	+	E10 Negative emotional states
2	E3 Uncertainty in the face of the pandemic	====˃˃	−	E14 Resilience against COVID-19
3	E3 Uncertainty in the face of the pandemic	====˃˃	+	E11 Suicide ideation
4	E10 negative emotional states	====˃˃	+	
5	E14 Resilience against COVID-19	====˃˃	−	
6	E3 Uncertainty in the face of the pandemic	====˃˃	+	E12 Drug addiction and alcoholism
7	E10 negative emotional states	====˃˃	+	
8	E14 Resilience against COVID-19	====˃˃	−	

Source: Own elaboration.

**Table 2 ijerph-18-12891-t002:** Expression of influence of the hypotheses.

Hypothesis	Expression
1	“E3 Uncertainty in the face of the pandemic influences in a direct significant way in E10 negative emotional states”.
2	“E3 Uncertainty in the face of the pandemic influences in an inverse significant way in E14 resilience against COVID-19”.
3	“E3 Uncertainty in the face of the pandemic influences in a direct significant way in E11 suicide ideation”.
4	“E10 Negative emotional states influence in a direct significant way in E11 suicide ideation”.
5	“E14 Resilience against COVID-19 influences in an inverse significant way in E11 suicide ideation”.
6	“E3 Uncertainty in the face of the pandemic influences in a direct significant way in E12 drug addiction and alcoholism”.
7	“E10 Negative emotional states influence in a direct significant way in E12 drug addiction and alcoholism”.
8	“E14 Resilience against COVID-19 influences in an inverse significant way in E12 drug addiction and alcoholism”.

Source: Own elaboration.

**Table 3 ijerph-18-12891-t003:** Descriptive statistics, reliability and validity of instruments.

Subscales	Mean	Standard Deviation	Cronbach’s Alpha	Rho_A	R-Squared	CR	AVE	E10	E11	E12	E3	E14
E10 Negative emotional states	2.53	0.78	0.73	0.74	0.220	0.85	0.65	0.81				
E11 Suicide ideation	1.22	0.48	0.83	0.86	0.173	0.88	0.65	0.306 **	0.81			
E12 Drug addiction and alcoholism	1.10	0.38	0.80	0.80	0.028	0.88	0.72	0.130 **	0.264 **	0.85		
E3 Uncertainty in the face of the pandemic	2.35	1.10	0.79	0.82	-	0.88	0.70	0.461 **	0.254 **	0.177 **	0.84	
E14 Resilience against COVID-19	3.66	0.94	0.85	0.85	0.017	0.90	0.68	−0.283 **	−3.36 **	−0.111 **	−0.127 **	0.83

** Significant correlation to level 0.01 (two-tailed); N = 5557; CR = composite reliability; AVE = average variance extracted; Main diagonal = square root of AVE. Source: own elaboration.

**Table 4 ijerph-18-12891-t004:** Student’s *t*-Test of mean differences between men and women.

Subscales	Sex	N	Mean	Deviation Dev.	Average Error Dev.	
E10 Negative emotional states	Women	3551	2.62	0.76	0.01	**
	Men	2006	2.38	0.79	0.02	
E11 Suicide ideation	Women	3551	1.23	0.48	0.01	
	Men	2006	1.22	0.48	0.01	
E12 Drug addiction and alcoholism	Women	3551	1.08	0.35	0.01	**
	Men	2006	1.12	0.43	0.01	
E14 Resilience against COVID-19	Women	3551	3.63	0.93	0.02	**
	Men	2006	3.71	0.94	0.02	
E3 Uncertainty in the face of the pandemic	Women	3551	2.49	1.11	0.02	**
	Men	2006	2.11	1.05	0.02	

** Significant differences at 95%. Source: own elaboration.

**Table 5 ijerph-18-12891-t005:** Student’s *t*-Test and Levene’s test of equal means.

Independent Samples Testing	Levene’s Test for Equality of Variances *t*-Test for Equality of Means										
Subscales		F	Sig.	t	DF	Two-Tailed Sig.	Mean Difference	SE Difference	95% of Confidence Interval of the Difference		
									Lower	Upper	
E10 Negative emotional states	Equal variances are assumed Equal variances are not assumed	3.292	0.07	10.953	5555.00	0	0.24	0.02	0.19	0.28	**
				10.858	4053.05	0	0.24	0.02	0.19	0.28	**
E11 Suicide ideation	Equal variances are assumed Equal variances are not assumed	0.012	0.912	0.738	5555.00	0.46	0.01	0.01	−0.02	0.04	
				0.738	4153.59	0.46	0.01	0.01	−0.02	0.04	
E12 Drug addiction and alcoholism	Equal variances are assumed Equal variances are not assumed	39.6	0	−3.329	5555.00	0	−0.04	0.01	−0.06	−0.01	**
				−3.159	3553.66	0	−0.04	0.01	−0.06	−0.01	**
E14 Resilience against COVID-19	Equal variances are assumed Equal variances are not assumed	0.34	0.56	−2.796	5555.00	0.01	−0.07	0.03	−0.12	−0.02	**
				−2.787	4120.29	0.01	−0.07	0.03	−0.12	−0.02	**
E3 Uncertainty in the face of the pandemic	Equal variances are assumed Equal variances are not assumed	17.748	0	12.377	5555.00	0	0.38	0.03	0.32	0.44	**
				12.598	4384.21	0	0.38	0.03	0.32	0.44	**

** Significant differences at 95%. Source: Own elaboration.

**Table 6 ijerph-18-12891-t006:** Bootstrapping of the structural equations modeling SEM PLS.

Hypothesis	Subscale	Original (O) Sample	Mean (M) of the Sample	Standard Deviation (Std Dev)	T Statistics (|O/Std Dev|)	*p* Values
H4	E10 Negative emotional states -> E11 Suicide ideation	0.152	0.151	0.013	11.341	0
H5	E14 Resilience against COVID-19 -> E11 Suicide ideation	−0.27	−0.27	0.014	18.947	0
H1	E3 Uncertainty in the face of the pandemic -> E10 Negative emotional states	0.47	0.469	0.01	45.374	0
H2	E3 Uncertainty in the face of the pandemic -> E11 Suicide ideation	0.15	0.151	0.016	9.586	0
H3	E3 Uncertainty in the face of the pandemic -> E14 Resilience against COVID-19	−0.13	−0.131	0.014	9.474	0

Source: own elaboration.

**Table 7 ijerph-18-12891-t007:** Summary of the influence of exogenous variables on endogenous variables in each of the hypotheses.

Hypothesis	Exogenous Variables	Influence	Standardized Beta Coefficient	Endogenous Variables	R Squared	Decision
1	E3 Uncertainty facing the pandemic	====>>	0.47	E10 Negative emotional states	0.2021	Accept
2	E3 Uncertainty facing the pandemic	====>>	−0.13	E14 Resilience against COVID-19	0.017	Accept
3	E3 Uncertainty facing the pandemic	====>>	0.15	E11 Suicide ideation	0.173	Accept
4	E10 Negative emotional states	====>>	0.152			Accept
5	E14 Resilience against COVID-19	====>>	−0.27			Accept

Source: own elaboration.

**Table 8 ijerph-18-12891-t008:** Increasing Mediation resulting values.

Hypotheses	Method	Path	Phat Coefficient	Indirect Effect	Standard dv	Total Effects	VAF	T	Sig.	*p*-Value	Decision
Incremental moderator effect/H1, H4, H3	Step 1. Direct effect (without mediation)	E3 ==˃E11	0.255	N/A	0.0128	0.255	N/A	19.86	***	0	Accept
	Step 2. Indirect effect (with mediation)	E3 ==˃E11	0.151	0.106	N/A		0.412	15.39	***	0	
		E3 ==˃E10	0.469		0.0069	0.257					
		E10 ==˃E11	0.226								

Source: own elaboration; *** significance at 95%.

**Table 9 ijerph-18-12891-t009:** Decreasing Mediation resulting values.

Hypotheses	Method	Path	Path Coefficient	Indirect Effects	Standard Dv	Total Effects	VAF	T	Sig.	*p*-Value	Decision
Decreasing moderating effect/H2, H3, H5	Step 1. Direct effect (without mediation)	E3 ==˃E11	0.255	N/A	0.013	0.255	N/A	19.864	**	0.000	Reject
	Step 2. Indirect effect (with mediation)	E3 ==˃E11	0.217	0.040	N/A		0.155	8.626	**	0.000	
		E3 ==˃E14	−0.132		0.005	0.257					
		E14 ==˃E11	−0.301								

Source: own elaboration; ** significative al 95%.

## Data Availability

The data presented in this study are available upon request.

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
