# Peer review of "Influence of COVID-19 Pandemic Uncertainty in Negative Emotional States and Resilience as Mediators against Suicide Ideation, Drug Addiction and Alcoholism"

_ijerph, 2021, doi:10.3390/ijerph182412891_

Round 1

Reviewer 1 Report

The authors presented an study evaluating the impact of uncertainty in different ourcomes related to emotional states and drug use, during COVID-19 pandemics. 

There are some reccommendations to authors:

  • The introduction is too long, and could be shortened.
  • In Methods section, point 2.2, describes the final sample, and should be placed in Results section.
  • Authors say that there was a 10% response rate; although the number of respondents is quite high (>5000), this low response rate is representative of degree students?
  • Among the instruments used, the second one described in the Methods, has been developed for this study. Is this instrument reliable? How has been designed and validated?
  • There are 2 Conclusions sections. The point 4 it is probably a Discussion section. This section, should be improved, as the 5th section (also conclusions).

Author Response

  1. from the introduction, we deleted paragraphs that were repetitive, shortening the introduction by 15 lines.
  2.  we moved point 2.2 to 3.1
  3. the population of this Campus is 11,000, therefore the sample corresponds to 45.45%, not 10% as stated by mistake.
  4. the instrument reliability and validity appear in the result section on point 3.3.
  5. point 4 was changed to discussion and it was improved.

Reviewer 2 Report

  1. On page 3 (line #112), the authors will need to pinpoint the “four basic pillars”. What are they?
  2. On page 7 (line #274), the authors reported that the response rate of the survey was less than 10%. If authors sent their survey to all students in the university, such response rate is very low. I assume that the authors did not use a random sampling or any sampling methods. Due to such low response rate, the authors will need to justify the representativeness of their study and limit of generalization of their findings. Furthermore, the authors will need to describe the distribution-and-collection process of the survey to all
  3. On page 7 (line #276), the first portion of statement “The survey consisted of five sections: questions regarding sociodemographic data and living conditions during …” appears to repeat itself in the remaining of the statement.
  4. The authors will need to clarify that they found a moderate correlation between uncertainty and negative emotional states, and correlations between uncertainty and other variables were weak. Furthermore, the authors need to explain the weak negative correlation between uncertainty and resilience. The discussion should be focused on the these two major findings. The implications will need to suggest some specific interventions to reduce uncertainty and increase resilience.
  5. On page 15 (line #511 to line #513; line #518 to #519), both of these suggested interventions are general and vague, they will need further elaboration and specification.
  6. Throughout the manuscript, the word “Anxiety” is mistakenly capitalized in the middle of many sentences.

Author Response

REVIEW 2 POINT-BY-POINT RESPONSE:

1. On page 3 (line #112), the authors will need to pinpoint the “four basic pillars”. What are they?

Response: The four basic pillars of resilience are described in lines 88-92.

2. On page 7 (line #274), the authors reported that the response rate of the survey was less than 10%. If authors sent their survey to all students in the university, such response rate is very low. I assume that the authors did not use a random sampling or any sampling methods. Due to such low response rate, the authors will need to justify the representativeness of their study and limit of generalization of their findings. Furthermore, the authors will need to describe the distribution-and-collection process of the survey to all.

Response: Line 245, point 2.2 data collection: the population of the campus is 10,975; therefore, the sample of 5,557 corresponds to 50.63%, not 10%, as stated by mistake.

3. On page 7 (line #276), the first portion of statement “The survey consisted of five sections: questions regarding sociodemographic data and living conditions during …” appears to repeat itself in the remaining of the statement.

Response: Page 7, line 276:  The paragraph related to the questionnaire sections was deleted because of repetitive information.

4. The authors will need to clarify that they found a moderate correlation between uncertainty and negative emotional states, and correlations between uncertainty and other variables were weak. Furthermore, the authors need to explain the weak negative correlation between uncertainty and resilience. The discussion should be focused on the these two major findings. The implications will need to suggest some specific interventions to reduce uncertainty and increase resilience.

Response: We improved the discussion section regarding the moderating effect of resilience on suicidal intention. All the information in the discussion and conclusion sections was grouped, and we deleted the conclusion section. 

5. On page 15 (line #511 to line #513; line #518 to #519), both of these suggested interventions are general and vague, they will need further elaboration and specification. 

Response: Interventions were explained in more detail in lines 478 to 485 and 488 to 492.

6. Throughout the manuscript, the word “Anxiety” is mistakenly capitalized in the middle of many sentences.

Response: The word Anxiety was decapitalized in all the text.

Round 2

Reviewer 1 Report

The authors have addressed the comments made in the previous review.

Reviewer 2 Report

The revisions are appropriate.